# A Protein Microarray-Based Respiratory Viral Antigen Testing Platform for COVID-19 Surveillance

**DOI:** 10.3390/biomedicines10092238

**Published:** 2022-09-09

**Authors:** Sungjun Beck, Rie Nakajima, Algis Jasinskas, Timothy J. Abram, Sun Jin Kim, Nader Bigdeli, Delia F. Tifrea, Jenny Hernandez-Davies, D. Huw Davies, Per Niklas Hedde, Philip L. Felgner, Weian Zhao

**Affiliations:** 1Department of Biological Chemistry, University of California, Irvine, CA 92697, USA; 2Department of Physiology and Biophysics, University of California, Irvine, CA 92697, USA; 3Velox Biosystems, 5 Mason, Suite 160, Irvine, CA 92618, USA; 4Department of Pharmaceutical Sciences, University of California, Irvine, CA 92697, USA; 5Student Health Center, University of California, Irvine, CA 92697, USA; 6Department of Pathology and Laboratory Medicine, University of California, Irvine, CA 92697, USA; 7Institute for Immunology, University of California, Irvine, CA 92697, USA; 8Laboratory for Fluorescence Dynamics, University of California, Irvine, CA 92697, USA; 9Beckman Laser Institute and Medical Clinic, University of California, Irvine, CA 92697, USA; 10Sue and Bill Gross Stem Cell Research Center, University of California, Irvine, CA 92697, USA; 11Chao Family Comprehensive Cancer Center, University of California, Irvine, CA 92697, USA; 12Edwards Life Sciences Center for Advanced Cardiovascular Technology, University of California, Irvine, CA 92697, USA; 13Department of Biomedical Engineering, University of California, Irvine, CA 92697, USA

**Keywords:** SARS-CoV-2, antibody microarray, antigen test, portable imager, immunofluorescence assay, tyramide signal amplification, protein biotinylation

## Abstract

High-throughput and rapid screening testing is highly desirable to effectively combat the rapidly evolving COVID-19 pandemic co-presents with influenza and seasonal common cold epidemics. Here, we present a general workflow for iterative development and validation of an antibody-based microarray assay for the detection of a respiratory viral panel: (a) antibody screening to quickly identify optimal reagents and assay conditions, (b) immunofluorescence assay design including signal amplification for low viral titers, (c) assay characterization with recombinant proteins, inactivated viral samples and clinical samples, and (d) multiplexing to detect a panel of common respiratory viruses. Using RT-PCR-confirmed SARS-CoV-2 positive and negative pharyngeal swab samples, we demonstrated that the antibody microarray assay exhibited a clinical sensitivity and specificity of 77.2% and 100%, respectively, which are comparable to existing FDA-authorized antigen tests. Moreover, the microarray assay is correlated with RT-PCR cycle threshold (Ct) values and is particularly effective in identifying high viral titers. The multiplexed assay can selectively detect SARS-CoV-2 and influenza virus, which can be used to discriminate these viral infections that share similar symptoms. Such protein microarray technology is amenable for scale-up and automation and can be broadly applied as a both diagnostic and research tool.

## 1. Introduction

The COVID-19 pandemic continues to spread widely and causes a significant impact on illness, death, economy, and everyday life. As the virus evolves continuously, as indicated by the recent Delta and Omicron variants, it is likely that the pandemic will continue in the conceivable near future. Screening tests play a critical role in COVID-19 surveillance, mitigation, and treatment intervention [1,2]. However, the field still lacks modular and scalable platforms that are amenable to rapid deployment to screen new viruses and variants. This was evident at the beginning of the COVID-19 pandemic as the shortage of robust tests significantly hindered our ability to constrain the spreading of infections.

In addition, only a few platforms are currently available to test a large number of different types of viruses (e.g., SARS-CoV-2 vs. influenza virus vs. common cold coronavirus) in a single assay, which can be important for physicians to accurately diagnose and treat respiratory diseases that share similar symptoms. Yet, significant efforts have been made in the research and development of biosensors and novel materials for both therapeutic and diagnostic purposes against SARS-CoV-2. For example, Rabiee et al. developed CaZnO-based nanoghost to detect SARS-CoV-spike antigen [3], Afshari et al. reviewed nanobubbles for hypoxic COVID-19 patients [4], and Ebrahimi et al. discussed graphene as a potential and promising material to develop novel molecular tests against pathogenic viruses, including SARS-CoV-2 [5]. Clinically, reverse transcriptase–polymerase chain reaction (RT-PCR) is however considered the current gold standard to detect SARS-CoV-2, which is typically run in centralized labs with sophisticated equipment, trained personnel, and with slow turnaround time and high cost. Therefore, RT-PCR limits its use as a routine and broad screening tool.

Various forms of antigen-based tests against SARS-CoV-2 [6,7] are available including point-of-care lateral flow assays (LFA) that are better suited for screening. For example, Giberti et al. examined the ND COVID-19 Ag test (NDFOS; Seoul, Republic of Korea), a lateral flow immunochromatographic test to detect both SARS-CoV-2 nucleocapsid protein (NP) and spike protein, using a total of 400 nasopharyngeal specimens, which confirmed its sensitivity of specificity as 85% and 100%, respectively [8]. Whereas, Thapa et al. developed a novel handheld thermo-photonic LFA reader, which enabled to quantify the LFA test results [9]. Above all, for these antigen tests to accurately capture antigen (or to neutralize pathogens as a therapeutic agent), high-quality monoclonal antibody is needed, and the quality of the antibody would determine the overall quality of the antigen test. Therefore, thorough characterization, testing, and screening of candidate monoclonal antibodies are critical (for a review, see [10]). Still, antigen tests are known to have lower sensitivity than PCR tests [11]. Therefore, one way to improve assay sensitivity is to target an abundant analyte. In the case of SARS-CoV-2, NP can be a suitable target due to its high abundance, compared to spike protein [12]. Furthermore, as these antigen tests typically rely on measuring only one or a small subset of antigens, they may be associated with a large false positive rate (poor specificity) due to cross-reactivity with seasonal human coronaviruses.

Protein microarrays represent a scalable, cost-effective, and potentially portable platform for parallel, multiplexed protein detection (for a review, see [13]), where each spot of an array can be composed of a specific monoclonal antibody to capture its target antigen at a stoichiometric ratio of 1:1. Hence, theoretically, the number of target antigen molecules should be equal to the number of specific monoclonal antibody molecules printed on protein microarrays. Using this technology, antigens from foodborne pathogens such as Escherichia coli O157:H7 [14] and toxins such as ricin and β-subunit of cholera toxin [15,16] were reported and detected. Specifically, the use of protein microarray as an in vitro diagnostic (IVD) platform is well reviewed in the literature [17]. Protein microarrays are typically prepared by printing proteins onto surface-modified microscope slides, which can be used to probe hundreds of different biomolecular binding events simultaneously in a single micro-sized test.

For instance, we have recently constructed a coronavirus antigen microarray (COVAM) containing 61 antigens that are responsible for acute respiratory infections including SARS-CoV-2 [18]. During the COVID-19 pandemic, the COVAM has served as a powerful tool for serological profiling to determine true COVID-19 prevalence, to identify suitable donors of convalescent plasma for transfusion treatment, and to investigate the breadth and durability of the immunologic response from vaccines [19]. In addition, to enable wide use of this microarray test, we also developed an inexpensive 3D-printable portable imaging platform (TinyArray imager) that can be deployed to read protein microarrays [20]. Based on COVAMs probed with COVID-19 positive sera, we have shown that the TinyArray imager produces concordant results to commercially available but expansive laser-based microarray imaging instruments [20].

In this study, we sought to adopt this protein-based microarray platform for robust, highly scalable, and multiplexed respiratory viral antigen analysis as a screening and diagnostic tool for COVID-19 surveillance. We printed antibodies on an array chip to probe multiple different antigens from common respiratory viruses including SARS-CoV-2, influenza virus, and seasonal human coronavirus. Printing antibodies are advantageous as it allows for to screen of numerous combinations of capture and detection antibodies for a given target antigen in a high-throughput manner, greatly reducing the time required to assemble a new panel against future variants and new pathogens. Here, we present the general workflow of validating the antibody microarrays for respiratory viral antigen detection including: (a) initial antibody reagent screening, (b) immunofluorescence assay design applying signal amplification approach (tyramide signal amplification, TSA) to detect low viral titers, (c) assay characterization with recombinant proteins, inactivated viral samples, and clinical samples, and (d) multiplexing for SARS-CoV-2 and influenza viruses. We believe this modular antibody microarray can potentially be deployed as a highly scalable, high-throughput, and multiplexed respiratory viral screening tool to address both the current COVID-19 pandemic and future outbreaks.

## 2. Materials and Methods

### 2.1. Recombinant Proteins and Antibodies

A summary of recombinant proteins and antibodies used in this study can be found in Appendix A.

### 2.2. Protein Biotinylation

To enable biotin-streptavidin interaction in our microarray-based immunoassays including multiplex antigen detection, some detection antibodies and recombinant proteins (reagent number 1, 3, 4, 5, 13, and 15 in Appendix A) were conjugated to biotin molecules using EZ-Link Sulfo-NHS-LC-Biotin kit (Thermo Fisher SCIENTIFIC, Waltham, MA, USA). Excess biotin molecules were removed using Zeba™ Spin Desalting Columns, 7K MWCO (Thermo Fisher SCIENTIFIC, Waltham, MA, USA) per the manufacturer’s protocol. To perform multiplex antigen testing, total protein biotinylation was performed on heat-inactivated SARS-CoV-2 culture fluid (Zeptometrix, Buffalo, NY, USA-WA1/2020 strain) and chemically inactivated influenza A H1N1 using the same EZ-Link Sulfo-NHS-LC-Biotin kit. Specifically, 100 μL of each inactivated virus sample was treated with 2 μL of 1 mM EZ-Link Sulfo-NHS-LC-Biotin for 30 min at room temperature on a rocker. Then, 10 μL of 10X TBST (G-Biosciences, St. Louis, MO, USA) was added and incubated for 30 min at room temperature on a rocker to quench the reaction. Excess biotin molecules were removed using Zeba™ Spin Desalting Columns, 7K MWCO (Thermo Fisher SCIENTIFIC, Waltham, MA, USA). The biotinylated total protein was immediately probed on an antibody microarray chip.

### 2.3. Inactivated SARS-CoV-2 Samples and SARS-CoV-2 Clinical Samples

Heat-inactivated SARS-CoV-2 culture fluid (Zeptometrix, Buffalo, NY, USA-WA1/2020 strain) was purchased and tested on the protein microarray as a positive control to characterize assay performance and normalize data for subsequent clinical sample analysis. RT-PCR-confirmed SARS-CoV-2 positive and negative samples were collected at the University of California Irvine (UCI) Institute for Clinical and Translational Services (ICTS) COVID Research Biobank, which is a biorepository for specimens from COVID-19 patients that has been approved by the Institutional Review Board of UCI (HS# 2012-8716). This study involved the use of de-identified biospecimens and a de-identified dataset, which is exempt from federal regulations under category 4ii as determined by the Institutional Review Board of UCI. Briefly, pharyngeal swabs were originally collected for diagnostic purposes using various RT-PCR assays. The swabs were maintained at 4 °C for a maximum 48 h until the diagnostic was set, then frozen at −80 °C. At the time of release for research, the clinical samples were thawed and aliquoted in universal transport media (UTM), MicroTest™ M4RT (Remel, San Diego, CA, USA) or Viral Transport Medium (HARDY, Santa Mari, CA, USA), in desired volumes. Inactivation by heating at 80 °C for 1 h [21] was performed, and the aliquots were kept at 4 °C and handed to the research laboratory within 24 h. Once the aliquots were received at the research laboratory, they were stored at −80 °C for long-term storage. These clinical positive samples were grouped in three different groups depending on the RT-PCR testing platforms utilized, which were all FDA-authorized. The Group 1 samples were tested by either Alinity m SARS-CoV-2 assay (Abbott Molecular Inc., Des Plaines, IL, USA) or Abbott RealTime SARS-CoV-2 assay (Abbott Molecular Inc., Des Plaines, IL, USA). The Group 2 samples were tested by either Simplexa COVID-19 Direct assay (DiaSorin Molecular LLC, Cypress, CA, USA), or ABI 7500 Fast Dx SARS-CoV-2 assay test on Thermo Fisher Applied Biosystems. The Group 3 samples were tested by either Xpert Xpress SARS-CoV-2 test (Cepheid, Sunnyvale, CA, USA) or Xpert Xpress CoV-2/Flu/RSV plus (Cepheid, Sunnyvale, CA, USA). E/(E + N2), S, N/N2, and RdRP genes or cycle threshold (Ct) only were reported and used as viral load assessment. Clinically validated negative control samples were also received from the UCI COVID Research Biobank following the same procedure described above. Both heat-inactivated SARS-CoV-2 culture fluid (Zeptometrix, Buffalo, NY, USA-WA1/2020 strain) and the SARS-CoV-2 clinical samples were diluted using the LY-13 buffer (ACROBiosystems, Newark, DE, USA) for probing.

### 2.4. Influenza A Virus Culture and Virus Titer Measurement

Reassortant influenza virus A/California/07/2009 (H1N1) x A/Puerto Rico/8/1934 was obtained from BEI Resources (BEI Resources, Manassas, VA, USA; NR-44004) and was grown in MDCK cells (ATCC, Manassas, VA, USA; CCL-34), following the culture method previously described in the literature [22]. Virus titer was determined using a microneutralization assay to determine a Tissue Culture Infectious Dose (TCID_50_/mL) in MDCK cells, as previously described in the literature [23]. The culture supernatant titer was determined to be 1 × 10^3^ TCID_50_/mL in MDCK cells. The 1 × 10^3^ TCID_50_/mL viral supernatant was then used to inoculate serum-pathogen-free (SPF) freshly fertilized eggs from Charles River Laboratories (Wilmington, MA, USA) as previously described in the literature [24]. The titer of virus in the allantoic fluid was determined to be 1 × 10^6^ TCID_50_/mL for the H1N1 via microneutralization assay and Reed-Muench method. The cultured influenza A virus was then chemically inactivated and lysed by diluting with the SafetyTector S Candor lysis buffer (CANDOR, Wangen, Germany).

### 2.5. Protein Microarray Printing

General protein microarray layout and design considerations were described in our previous coronavirus antigen microarray (COVAM) study [18]. Briefly, before printing proteins, the concentration of recombinant proteins and antibodies was measured by either Qubit Protein Assay Kit (Thermo Fisher SCIENTIFIC, Waltham, MA, USA) or NanoDrop™ spectrophotometers by measuring A280 (Thermo Fisher SCIENTIFIC, Waltham, MA, USA). Antibody microarray fabrication used in this study is principally the same as described previously [25] with some modifications addressed in the literature [26]. Briefly, proteins were printed onto ONCYTE AVID nitrocellulose-coated glass slides (GRACE BIO-LABS, Bend, OR, USA) using an OmniGrid 100 microarray printer (GeneMachines, San Carlos, CA, USA). Each array slide contains 2 × 8 pads, which allows 16 different samples to be analyzed per slide. The printed protein microarrays were stored in a vacuum desiccator at room temperature until use.

### 2.6. Fluorescence Signal Development and Reagent Screening

Before probing samples, FAST™ Protein Array Blocking Buffer (GVS, Zola Predosa, Italy) was used to rehydrate nitrocellulose pads. Detection antibodies, secondary antibodies, and streptavidin-fluorophore conjugates were all prepared in the same blocking buffer. Two streptavidin-fluorophore conjugates were used in this study to develop fluorescence signals: Alexa Fluor^®^ 488 Streptavidin (Jackson ImmunoResearch, West Grove, PA, USA) at a final concentration of 1.7 mg/mL, and Streptavidin, Alexa Fluor™ 647 Conjugate (Thermo Fisher SCIENTIFIC, Waltham, MA, USA), at a final concentration of 2 mg/mL. All fluorescence signals were developed using the streptavidin-Alexa Fluor™ 647 conjugate unless specified. The same working volume and dilution factor of 100 μL and 250 were used. Initially, four lysis buffers were tested to prepare the dilution of the heat-inactivated SARS-CoV-2 (Zeptometrix, Buffalo, NY, USA-WA1/2020 strain): 1X PBST (G-Biosciences, St. Louis, MO, USA), LY-13 lysis buffer (ACROBiosystems, Newark, DE, USA), SafetyTector S Candor lysis buffer (CANDOR, Wangen, Germany), and Quidel extraction buffer (Quidel, San Diego, CA, USA). The LY-13 buffer was determined to yield an optimal signal-to-noise ratio with the least variation and therefore used subsequently for assaying heat-inactivated SARS-CoV-2 and all clinical samples.

### 2.7. Antibody Pair Screening for Sandwich Immunoassays on Protein Microarrays

Recombinant viral proteins were diluted by five-fold serial dilutions, starting from either 217 nM or 792 nM for SARS-CoV-2 NP (reagent number 1 in Appendix A) and 376 nM for Spike RBD (reagent number 2 in Appendix A) in FAST™ Protein Array Blocking Buffer (GVS, Zola Predosa, Italy) (Appendix A). The diluted recombinant viral proteins were probed on a protein microarray slide to screen both capture and detection antibodies for specific SARS-CoV-2 antigens. The information was used to estimate K_D_ and Bmax. ProPlate^®^ Multi-Well Chambers (GRACE BIO-LABS, Bend, OR, USA) were placed over the microarray slide and secured with clips, then slide/chamber assemblies were positioned into a ProPlate^®^ Tray (GRACE BIO-LABS, Bend, OR, USA). Each pad on the protein microarray was hydrated using the blocking buffer. After removing the blocking buffer, the arrays were probed with pre-diluted recombinant viral proteins for 1 h at room temperature with gentle agitation. The arrays were then washed three times with 1X TBST (G-Biosciences, St. Louis, MO, USA). The same probing method was applied for the detection of antibodies and secondary antibodies. The streptavidin-fluorophore conjugates were probed for 30 min at room temperature with gentle agitation. After probing and washing, all slides were dried by centrifuging at 500× *g* for 5 min before imaging.

### 2.8. Clinical SARS-CoV-2 Samples and Multiplex Viral Antigen Detection

For the inactivated viral sample and clinical sample testing (Figures 3 and 4), the probing volume was 100 μL. All samples were diluted to 1:5 using the LY-13 lysis buffer. For the recombinant viral protein detection experiments (Figure 5), the probing volume was 50 μL, where biotinylated recombinant viral proteins were diluted in the blocking buffer to achieve the final probing concentrations of 0.005 mg/mL, 0.0025 mg/mL, and 0.00125 mg/mL. For the multiplex virus detection via total protein biotinylation of the inactivated virus samples (Figure 6), the probing volume was also 50 μL, where 1X PBST (G-Biosciences, St. Louis, MO, USA) and SafetyTector S Candor lysis buffer (CANDOR, Wangen, Germany) were used to lyse heat-inactivated SARS-CoV-2 (Zeptometrix, Buffalo, NY, USA-WA1/2020 strain) and chemically inactivated influenza A H1N1, respectively. After the total protein biotinylation, these samples were desalted by using Zeba columns 7K MWCO (Thermo Fisher SCIENTIFIC, Waltham, MA, USA). For positive controls, biotinylated recombinant SARS-CoV-2 NP and influenza A H1N1 full-length HA0 (reagent number 1 and 3 in Appendix A, respectively) diluted in the blocking buffer were probed. Samples were incubated with array pads for 1 h with gentle agitation on a rocker. Detection antibodies, secondary antibodies, and streptavidin-fluorophore conjugates were subsequently added and incubated for 1 h, 1 h, and 30 min, respectively. All incubation steps were performed at room temperature. Typical antibody concentrations used for this experiment were provided in Appendix A. All antibodies were diluted at the dilution factor of 200–250 with a working volume of 100 μL. After each step, 100 μL 1X TBST (G-Biosciences, St. Louis, MO, USA) was used to wash the pads. To amplify the fluorescence signals on the protein microarray, Alexa Fluor™ 488 Tyramide SuperBoost™ Kit, streptavidin (Thermo Fisher SCIENTIFIC, Waltham, MA, USA) was used, following the manufacturer’s protocol. After probing and washing, all slides were dried by centrifuging at 500× *g* for 5 min before imaging.

### 2.9. Protein Microarray Imaging and Data Analysis

Protein microarray imaging and data analysis. To read the microarray fluorescence, a refined version of our microarray TinyArray imager was used [20] unless specified (the antibody pairs 4, 5, and 6 in Appendix A were tested and imaged using the Perkin Elmer ScanArray Express HT confocal laser scanner (Perkin Elmer, Waltham, MA, USA) at a wavelength of 670 nm). To excite Alexa Fluor™ 647 fluorescence, a red LED (625 nm emission peak) was collimated with a lens of 20 mm focal length and spectrally cleaned with a band pass filter (ET630/20x, Chroma, Bellows Falls, VT, USA) to homogeneously illuminate the sample within an area of ~20 × 20 mm^2^ that covered 2 × 2 microarray pads. To excite Alexa Fluor™ 488 fluorescence, a blue LED (490 nm emission peak) and corresponding band pass filter (ET460/30x, Chroma, Bellows Falls, VT, USA) were used. A servo motor was used to translate the slide to capture a total of four images to quantify the entire 16-pad microarray slide. The fluorescence signal of Alexa Fluor™ 647 or Alexa Fluor™ 488 was separated from excitation light with suitable band pass filters (AT690/50m or ET520/20m, Chroma, Bellows Falls, VT, USA) and imaged with a camera lens of 100 mm focal length (Edmund Optics, Barrington, NJ, USA) onto the chip of a 20-megapixel monochrome camera (BFS-U3-200S6M-C, FLIR, Thousand Oaks, CA, USA). Camera gain was 0 dB and the exposure time was 2500 ms for the Alexa Fluor™ 647 fluorescence and 500 ms for the Alexa Fluor™ 488 fluorescence. Acquired microarray images were analyzed using commercially available ProScanArray Express software (Perkin Elmer, Waltham, MA, USA). Each spot on the pads was quantified in median values. All signal intensities were corrected for spot-specific local backgrounds. Negative relative fluorescence units (RFU) were converted to zero, and raw RFU values were reported in the figures, except for the clinical sample data. For clinical sample data analysis, to address inter-assay variability, fluorescence intensity (in RFU) was normalized using an internal positive control, namely, RFU from the heat-inactivated SARS-CoV-2 culture fluid (Zeptometrix, Buffalo, NY, USA-WA1/2020 strain) probed on the same protein microarray slide. The normalization was performed by taking ratios between the mean RFU of the internal positive control and individual RFUs of the clinical samples. 

### 2.10. Statistical Methods

To achieve K_D_ and Bmax of the antibody pairs against recombinant SARS-CoV-2 antigens (Appendix A), curve-fitting was performed using a built-in function in PRISM software (GraphPad, San Diego, CA, USA): nonlinear regression and one site-specific binding analysis. To calculate the limit of detection (LoD) against the heat-inactivated SARS-CoV-2 culture fluid (Zeptometrix, Buffalo, NY, USA-WA1/2020 strain) using the optimal immunoassay condition, linear regression was performed using PRISM software (GraphPad, San Diego, CA, USA) (Figure 3). For clinical sample data (Figure 4), an identical negative clinical sample (sample ID 19040) was used to determine thresholds to distinguish true positives and false positives from the 22 RT-PCR-confirmed positively diagnosed SASR-CoV-2 clinical samples on the protein microarray. The threshold was calculated by measuring the mean of the clinical negative sample (sample ID 19040) plus two or three standard deviations (T = X + 2S or 3S, where T = the threshold, X = the mean, and S = the standard deviation [27]). The lenient threshold was calculated with 2S, and the stringent threshold was calculated with 3S. To test the clinical specificity, a total of five clinical negative samples were tested and compared to the no antigen control (Figure S3). One-way ANOVA, followed by Tukey’s multiple comparisons, was performed using the built-in function in PRISM software (GraphPad, San Diego, CA, USA). To compare the multiplex virus detection data (Figure 6), One-way ANOVA, followed by Tukey’s multiple comparisons, was also performed using the built-in function in PRISM software (GraphPad, San Diego, CA, USA). Descriptive figures such as illustrations were designed and created with BioRender.com (BioRender, Toronto, ON, Canada). Graphs were generated by using either PRISM software (GraphPad, San Diego, CA, USA) or Microsoft Excel (Microsoft, Redmond, WA, USA).

## 3. Results

### 3.1. Protein Microarray Design Considerations and Assay Workflow for Viral Antigen Detection

Our goal was to develop an antibody microarray for multiplexed respiratory viral testing, comprising numerous immobilized antibodies against viral antigens from most common respiratory viruses, such as SARS-CoV-2, influenza viruses, and seasonal human coronavirus strains. The general assay workflow is shown in Figure 1a. Independent spots on the microarray were first printed using an automated robotic contact printer with capture antibodies specific to target viral antigens. To quality control, the slide batch and printing, 80 spots of biotin-BSA at a concentration of 0.1 mg/mL were deposited on each pad of 16-pad slides, and then slides were developed with Streptavidin- Alexa Fluor™ 647 conjugates. After signal quantitation, the coefficient of variation (CV) was calculated for each pad and for the slide. Commonly, CV was not exceeding 20% (data not shown). For a particular antibody array experiment, CV was calculated after slide probing using deposited human IgG as control and was not exceeding 20% as well (data not shown). Samples were added and probed applying sandwich immunoassay methods using detection antibodies and fluorescent tags for single-target detection (Figure 1b–d) and applying a non-sandwich method, total protein biotinylation approach, for multiplexed detection (Figure 1e).

Sequential sandwich immunoassay with both capture and detection antibodies ensures target detection specificity as it requires detection antibodies that bind to a different epitope of the same antigen [28]. A fluorescence signal was developed using biotinylated secondary antibodies complexed with streptavidin-fluorophore conjugates (Figure 1b). In some cases, biotinylated detection antibodies were used and complexed with streptavidin-fluorophore conjugates (Figure 1c). To further improve assay sensitivity, we also applied the signal amplification method tyramide signal amplification (TSA) (Figure 1d), when probing clinical samples with low viral titers. Through TSA, activated tyramide-fluorophore conjugates (Alexa Fluor^®^ 488 in this study) bind to tyrosine residues of proteins, labeling a large number of fluorophores on the immunocomplex [29,30]. For multiplexed testing, total viral proteins are biotinylated, captured by the corresponding capture antibody on the array, and then probed using streptavidin-fluorophore to develop fluorescence signals (Figure 1e). This detection scheme was chosen for multiplexing detection because it can avoid potential cross-reactivity between capture antibodies and detection antibodies for different targets. However, we understand protein biotinylation of a single clinical sample would be expensive, time-consuming, and extra laborious. In addition, such detection would not be able to provide a quantitative measure. However, one advantage of this strategy would be that multiple clinical samples may be pooled for total protein biotinylation to perform a community-level general diagnostic test to improve the throughput of the diagnostic test [31], providing information for rapid decision-making processes in a pandemic or local epidemics. After probing samples, images were acquired using our portable TinyArray imager [20], unless specified. Microarray fluorescence data were analyzed using ProScanArray Express software as described in Materials and Methods.

The modular design of protein (antibody) microarray allowed us to conduct many iterations quickly, with each iteration designed for a particular purpose such as screening for optimal capture and detection antibody pairs, buffer conditions, or characterizing antigen dose–response with capture antibodies serially diluted on the microarray. Our initial version of antibody microarrays was focused on characterizing SARS-CoV-2 antigen detection whereas the final version further incorporated capture antibodies to detect influenza viruses and a common cold coronavirus. In addition, fiducial alignment markers consisting of fluorescent dye were printed in a specific pattern to assist with automated spot profiling in data analysis. Internal controls including human IgG and mouse IgG were also printed. In some iterations, we deposited a concentration gradient of capture antibodies each across several orders of magnitude which served as an internal calibration control for quantification and normalization purposes for RT-PCR-confirmed clinical samples. Each capture antibody was printed at least in duplicates. General considerations of protein microarray construction were described in our previous COVAM study [18]. In the following sections, we will only present key representative iterations that illustrate the development process of the antibody microarray and the final versions that were used for clinical samples and multiplexed respiratory viral antigen detection.

**Figure 1 biomedicines-10-02238-f001:**
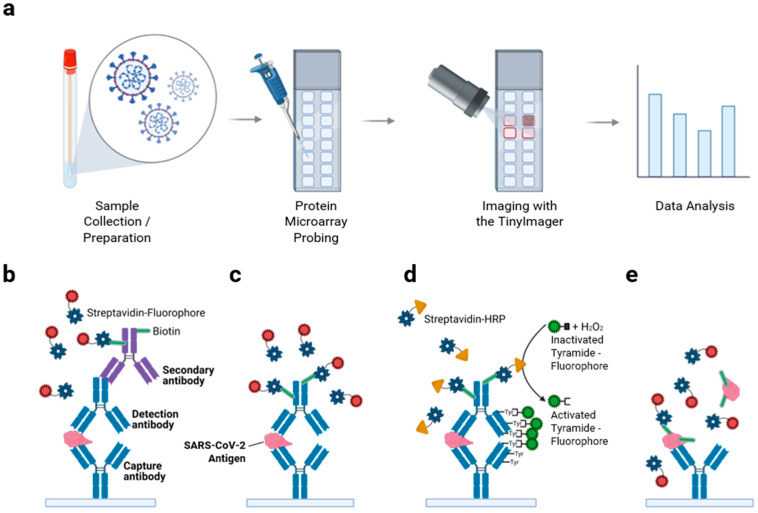
Antibody microarray workflow and antigen detection scheme. (**a**) An illustration to describe the workflow of a protein (antibody) microarray chip is shown. Clinical samples, inactivated viruses, and recombinant viral proteins can be probed on a protein microarray chip. In this study, the microarray images were acquired by the TinyArray imager [20] unless otherwise specified. Fluorescence data analysis was performed using ProScanArray Express software. Single antigen detection schemes used in this study are shown in (**b**–**d**). In (**b**), a detection antibody, biotinylated host-specific secondary antibody, and streptavidin-fluorophore conjugates were used to develop fluorescence signals. Streptavidin was illustrated in its tetramer form. Biotin molecules were illustrated as green rods. In (**c**), biotinylated detection antibodies and streptavidin-fluorophore conjugates were used to develop fluorescence signals. In (**d**), biotinylated detection antibody and streptavidin-horseradish peroxidase (HRP) conjugates, illustrated as yellow triangles, were used to perform downstream tyramide signal amplification (TSA) to amplify fluorescence signal. (**e**) For multiplexed testing of different targets, total viral proteins are biotinylated and captured by corresponding antibodies on the array. Then, streptavidin-fluorophore conjugates were probed to develop fluorescence signals. This figure was created with BioRender.com.

### 3.2. Screening for Optimal Anti-SARS-CoV-2 Antibodies

The performance of an antigen test is critically dependent on the quality of antibodies. An advantage of the microarray platform is that it can screen numerous reagents from different vendors (in this case combinations of capture and detection antibodies) and experimental conditions (e.g., detection schemes) in a high-throughput manner, greatly reducing the time for assay development. Specifically, to identify the optimal antibody pair against SARS-CoV-2 antigens, we screened multiples of commercially available antibody pairs, where Appendix A summarizes the details of antibody information, experiment conditions, and results. While our early microarray iterations contained antibodies targeting various SARS-CoV-2 antigens including spike proteins (a representative data shown as the pair number 6 for the spike protein in Appendix A), we later focused on NP as the main antigen of interest for SARS-CoV-2 due to its high abundance [12]; indeed, NP has been a commonly used target for COVID antigen test thus far [32,33,34]. Briefly, to test each antibody pair (Appendix A), serially diluted (typically diluted five-fold for ten concentration points) SARS-CoV-2 recombinant antigens were probed following the detection schemes shown in Figure 1b or Figure 1c. 

Figure 2 demonstrated a representative microarray slide image, a pad image with protein arrays, and the antigen titration curve obtained with the antibody pair number 1 in Appendix A for SARS-CoV-2 NP antigen. This type of sandwich-based immunoassay typically follows a sigmoidal curve [35,36]. Therefore, we fit antigen titration data to nonlinear regression using a built-in function in PRISM software (One site—Specific binding). After curve-fitting, we found some antibody pairs showed a decreasing trend in RFUs from high analyte concentration points (Figure 2c). Such a trend can be explained by the high-dose hook effect, where too much analyte saturated both capture and detection antibodies, yielding low signals [35]. From each curve, we examined two parameters: Bmax and dissociate constant (K_D_), where Bmax represents the maximum specific binding (represented in RFU), and K_D_ represents a specific antigen (or antigen-antibody) concentration to achieve equilibrium at half of the maximum binding. Thus, the lower the K_D_ is, the stronger the binding is between the antigen and the antibody. The K_D_ in this study represented the binding between the capture antibody and the rest of the immunocomplex, that is, the biotinylated detection antibody (together with biotinylated secondary antibodies if used) and streptavidin-fluorophore conjugates as a whole. Through this screening experiment, we determined that pair number 1 (Capture antibody: ACROBiosystems NUN-S46, Detection antibody: ACROBiosystems NUN-S47) was the optimal pair for recombinant NP antigen detection, resulting in the lowest K_D_ (0.733 nM) and relatively short assay time (2.5 h). As our assay design follows a typical fluorescence-linked immunosorbent assay (FLISA) format, most of the assay time results from incubation time for each step (see Section 2 for more details).

### 3.3. Antibody Microarray Characterization Using Inactivated SARS-CoV-2 Samples

We next characterized the SARS-CoV-2 antibody microarray using heat-inactivated SARS-CoV-2 culture fluid samples. As SARS-CoV-2 NP is found inside of the virion [37], for effective antigen-based testing, we would need an appropriate lysis buffer to dissociate the virion to make NP accessible. Four lysis buffers including 1X PBST (Tween-20 0.05% *v*/*v*), LY-13, Candor Lysis Buffer (SafetyTector S), and Quidel extraction buffer were tested. The choice of detergent is especially important because a high concentration of strong ionic detergent such as sodium dodecyl sulfate (SDS) may interrupt antibody-antigen interaction after protein denaturation [38]. For this set of experiments, the same capture antibody (ACROBiosystems NUN-S46, reagent number 14 Appendix A from the antibody pair 1, Appendix A) was used for detection. Besides the detection antibody from antibody pair 1, Appendix A (ACROBiosystems NUN-S47, reagent number 15 Appendix A), during our iterative development process, another detection antibody (Creative Diagnostics CABT-RMJ1, reagent number 13 Appendix A) became commercially available which was specifically reported to be suitable for NP antigen test from viral samples, so we tested it as well. Appendix A summarizes the SARS-CoV-2 NP detection results from a combination of the four lysis buffers and the two detection antibodies.

Commercially purchased heat-inactivated SARS-CoV-2 sample was diluted ten-fold, up to the final concentration of 1.15 × 10^6^ TCID_50_/mL, with each lysis buffer and probed with two different detection antibodies: CABT-RMJ1 and NUN-S47. The two detection antibodies were diluted using the same dilution factor of 250 from the neat concentration and probed with the same volume of 50 μL. The CABT-RMJ1 had a higher stock concentration than the NUN-S47, which may explain why it yielded higher RFUs (Appendix A). The commercial lysis buffer LY-13 yielded the highest signal-to-noise ratio (SNR) with the least variation, which is about an SNR of 12 when used with the CABT-RMJ1 detection antibody. Interestingly, we found 1X PBST also yielded a decent SNR, which was about an SNR of 9, with the CABT-RMJ1 detection antibody. This suggests that a nonionic detergent such as Tween-20 would be sufficient to release SARS-CoV-2 NP. Nevertheless, we chose LY-13 for our following assay development. In summary, we determined the following reagents to be optimal for SARS-CoV-2 antigen testing with complex viral samples: NUN-S46 capture antibody, CABT-RMJ1 detection antibody, and LY-13 lysis buffer, which were used for subsequent experiments.

Using these optimal reagents, we characterized analytical performance of the protein microarray against the same heat-inactivated SARS-CoV-2 sample. For this experiment, we printed a concentration gradient of the capture antibody on the microarray (a total of 8 concentration points ranging from 0.009 mg/mL to 1.200 mg/mL). The inactivated SARS-CoV-2 sample was serially diluted with the LY-13 lysis buffer in a two-fold and five-fold manner, probing 7 and 14 concentration points, respectively. Figure 3a shows representative microarray images after probing heat-inactivated SARS-CoV-2 samples, demonstrating a dose-dependent response per both capture antibody and analyte concentrations. Fluorescence images were analyzed, linear regression was performed, and then the limit of detection (LoD) was calculated. Figure 3b represents linear regression between the inactivated SARS-CoV-2 concentration and the RFU. The LoD was calculated rather stringently by multiplying 3.3 to the standard deviation of the y-intercept over the slope, which was determined to be 1.54 × 10^5^ TCID_50_/mL. This LoD is generally comparable to those obtained from other SARS-CoV-2 antigen tests [39,40,41]. In addition, the heat inactivation of the SARS-CoV-2 samples decreases the virus titer up to 4 Log10 in TCID_50_ [42], which affected the antigen detection on the protein microarray and therefore the LoD calculation. Furthermore, with two independent experiments, the microarray assay demonstrated acceptable precision and repeatability with coefficient of variation (CV) % ranging from 1.14% to 36.14% (virus titer from 9.20 × 10^4^ TCID_50_/mL to 2.30 × 10^6^ TCID_50_/mL Appendix A). The CV% drastically increased near the calculated LoD value, which was expected.

**Figure 3 biomedicines-10-02238-f003:**
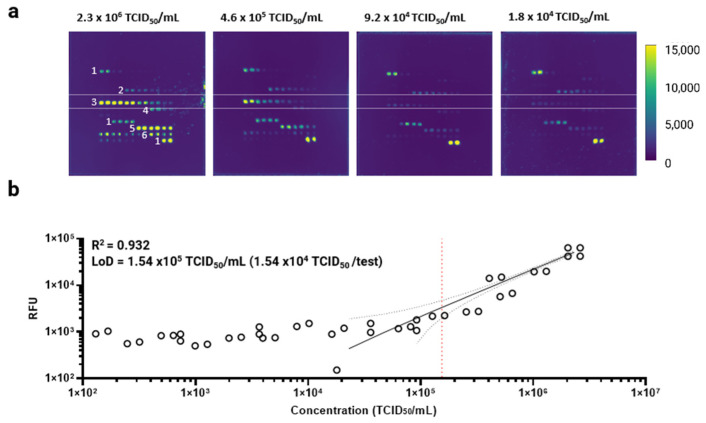
Characterization of the protein microarray against heat-inactivated SARS-CoV-2 culture fluid. Heat-inactivated SARS-CoV-2 sample was serially diluted two-fold and five-fold in two independent titration experiments, respectively. (**a**) Example microarray images representing fluorescence signals developed from different virus titers are shown. The two white lines (label 3) indicate the location of the optimal anti-SARS-CoV-2 NP capture antibody printed at different concentrations. (**b**) Using the highest capture antibody concentration in duplicates, linear regression was performed between the virus titer in TCID_50_/mL and raw RFU. The data were plotted on a log–log scale. The black straight line represents the linear regression line. The red dotted line represents the calculated limit of detection (LoD), which was calculated by multiplying 3.3 by a standard deviation of the y-intercept over the slope. The black dotted line represents the 95% confidence band of the linear regression line.

### 3.4. Antibody Microarray Test Evaluation with Clinical Samples

To evaluate its potential clinical utility, we next characterized the antibody microarray using clinically confirmed, SARS-CoV-2 RT-PCR positive (total of 22) and negative (total of 5) patient samples. These positive clinical samples were previously tested with FDA-authorized SARS-CoV-2 RT-PCR tests. Clinical sample information including cycle threshold (Ct) value is shown in Appendix A. These samples exhibited a broad range of Ct values, which allowed us to potentially correlate our microarray assay performance with viral titers. Using the optimal capture and detection antibody identified above and detection scheme shown in Figure 1c, we first demonstrated that all RT-PCR negative clinical samples were negative compared to the background signal (clinical specificity of 100.0%, Appendix A) on the protein microarray.

To assess clinical sensitivity for clinically confirmed positive cases, we first established our thresholding method. First, to address potential inter-assay variability, all clinical sample fluorescence data were normalized using an internal positive control (i.e., the heat-inactivated SARS-CoV-2 culture fluid sample) probed on the same array. We then set the threshold with which we determine if a clinical sample is positive or negative by measuring the mean of multiple replicates (*n* = 8 per experiment) of an identical negative clinical sample (sample ID 19040), plus three standard deviations. Using the stringent three standard deviations threshold, we found 11 out 22 RT-PCR positive clinical samples were positive using our microarray assay (clinical sensitivity of 50.0%) (Figure 4a,c,e). Furthermore, we noticed a trend that our microarray fluorescence signal (normalized RFU) generally correlates with the viral titers in terms of the Ct values (with the only exception of sample ID 18812 where target antigens might have been potentially degraded post RT-PCR sample processing). The positive samples based on microarray data are typically those that exhibited low Ct values from RT-PCR (e.g., Ct values were 17.66 for sample ID 18826 in Group 1, 15.90 for sample ID 18879 in Group 2, and 20.10 for sample ID 18834 in Group 3, respectively). Granted, each RT-PCR kit may target different viral RNA sequences, and direct comparison between the Ct values among different RT-PCR test groups can be challenging to understand true virus titers and its correlation to the microarray-based SARS-CoV-2 NP level in each sample.

To further improve clinical sensitivity for the clinical samples with low viral titers, we applied tyramide signal amplification (TSA) that can label a large number of fluorophores on the bound immunocomplex on the array. Following TSA reaction, we observed a significant increase in fluorescence intensity on the microarray (Appendix A). Since TSA amplifies signal rather non-specifically, it is important to optimize reagent incubation time to achieve a high SNR. By probing recombinant SARS-CoV-2 NP with different TSA reagent incubation time, we determined two-minute incubation yielded optimal SNR (Appendix A), which was therefore used subsequently for clinical samples. With TSA, we re-tested the low titer (high Ct values) “negative” clinical samples from Figure 4a,c,e. Six low titer samples were determined to be positive when the lenient two standard deviations threshold (Threshold 2) was used (Figure 4b,d,f); we determined the test result of each low titer sample by comparing the mean normalized RFU value from the two technical replicates to the Threshold 2. In the case of the Group 2 low titer samples (Figure 4d, sample ID 18829, 18913, and 18959), we observed some variations in the two technical replicates, which could have been resulted from printing errors, possibly from preparation of this particular microarray chip. Nevertheless, this led to an improved clinical sensitivity of 77.2% (17 out of 22). Accordingly, the results indicate the clinical sensitivity of an antigen test can be highly dependent on the virus titer, where the risk of bias from participant or sample selection should be considered while examining and developing an antigen test [7].

**Figure 4 biomedicines-10-02238-f004:**
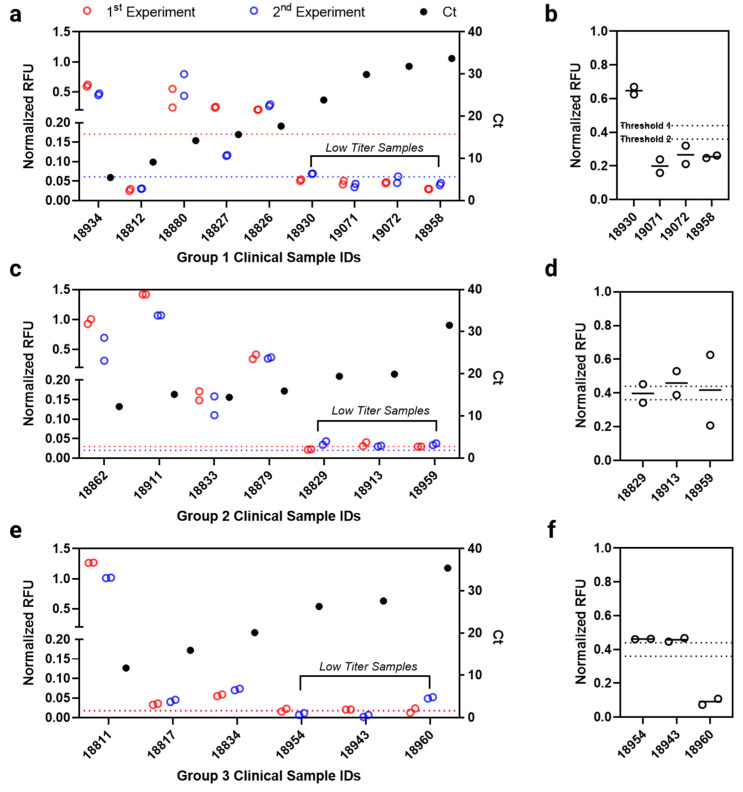
Clinical sample testing on the protein microarray. A total of 22 SARS-CoV-2 clinical samples was tested on the protein microarrays. These clinically diagnosed positive samples were previously tested by one of three FDA-authorized RT-PCR kit combinations: (**a**) Group 1 Alinity m SARS-CoV-2 assay or Abbott RealTime SARS-CoV-2 assay, (**c**) Group 2 Simplexa COVID-19 Direct assay or ABI 7500 Fast Dx SARS-CoV-2 assay test on the Thermo Fisher Applied Biosystems, (**e**) Group 3 Xpert Xpress SARS-CoV-2 test or Xpert Xpress CoV-2/Flu/RSV plus. Fluorescence signals were developed by following the scheme represented in the Figure 1c. Two independent experiments (a total of four technical replicates) were performed per sample for each group. Each clinical sample’s individual normalized RFU value is reported. The cycle threshold (Ct) value of each clinical sample is also reported on the right Y-axis. The threshold was calculated by adding average of the normalized RFU value of the negatively diagnosed clinical sample (*n* = 8 technical replicates) to its 3X standard deviation from all experiments. These thresholds were indicated by dotted lines on each graph. The samples below (or almost near) the threshold were designated as the “low titer samples”. (**b**,**d**,**f**) The low titer samples were re-tested with tyramide signal amplification (TSA), following the detection scheme represented in the Figure 1d. A single experiment with two technical replicates were performed where each clinical sample’s individual normalized RFU values are reported in hollow circles and their mean values as horizontal lines. The thresholds were calculated in the same manner where the Threshold 1 and 2 used 3X and 2X standard deviation of the negatively diagnosed clinical samples, respectively.

We further noted that, due to safety concerns, we were not able to collect fresh swabs from patients in this study. Rather, our clinical swab samples were aliquoted in UTM and then heat-inactivated before use. Therefore, there was a sample dilution prior to analysis which impaired the sensitivity. In addition, inactivation method can potentially affect protein and epitope integrity, thereby interfering antibody-antigen binding. We expect for real-world screening and diagnostic applications, where we can collect dry swab samples for immediate probing in the lysis buffer without UTM dilution, we will be able to further improve clinical sensitivity which can readily meet the minimal performance criteria for SARS-CoV-2 antigen tests set by regulatory bodies [43]. Nevertheless, our current clinical sensitivity (77.2%) and specificity (100.0%) are generally comparable to existing COVID-19 antigen tests [34,44].

### 3.5. Multiplexed Antibody Microarray Test for Respiratory Viruses

Finally, we constructed a prototype antibody microarray chip to test multiplexed detection of several most common respiratory viruses including SARS-CoV-2, seasonal common cold coronavirus, influenza A virus, and influenza B virus. Specifically, this microarray comprises four different capture antibodies printed on each pad that detect antigens of SARS-CoV-2 (NP, as established above), OC43 common cold coronavirus (OC43 NP), influenza A virus (influenza A hemagglutinin H1N1 A/California/04/2009 HA0), and influenza B virus (influenza B/Florida/4/2006 HA0). The capture antibody information is provided in Appendix A. Again, for coronaviruses, NP was chosen as the main target antigen due to its high abundance. For influenza virus, HA was used as the target antigen due to its functionality as a receptor binding protein, high abundance, and specificity to seasonal strains [45,46,47].

Here, we defined the term singleplex as probing each individual recombinant viral protein after biotinylation and the term multiplex as probing whole virus lysate after biotinylation, probing multiple viral proteins. For multiplex detection, we used a non-sandwich immunoassay where total viral proteins are biotinylated, captured by corresponding antibodies on the array, and then probed using streptavidin-fluorophore conjugates to develop fluorescence signals (Figure 1e). This scheme can avoid potential cross-reactivity between capture antibodies and detection antibodies among different targets as typically found in sandwich-based assays, but assay specificity can be compromised because a single epitope is targeted. Therefore, to test the assay specificity using the detection scheme described by Figure 1e, four recombinant viral target antigens were individually biotinylated and probed at three different concentrations including 0.005 mg/mL, 0.0025 mg/mL, and 0.00125 mg/mL as singleplex detections. Fluorescence signal of each target (RFU) generally follows a dose–response manner with the exception of the influenza B HA, where its fluorescence signal was almost saturated for all three tested concentrations (Figure 5). Nevertheless, the antibody microarray assay showed great specificity among four different targets with nearly no noticeable cross-reactivity (Figure 5).

**Figure 5 biomedicines-10-02238-f005:**
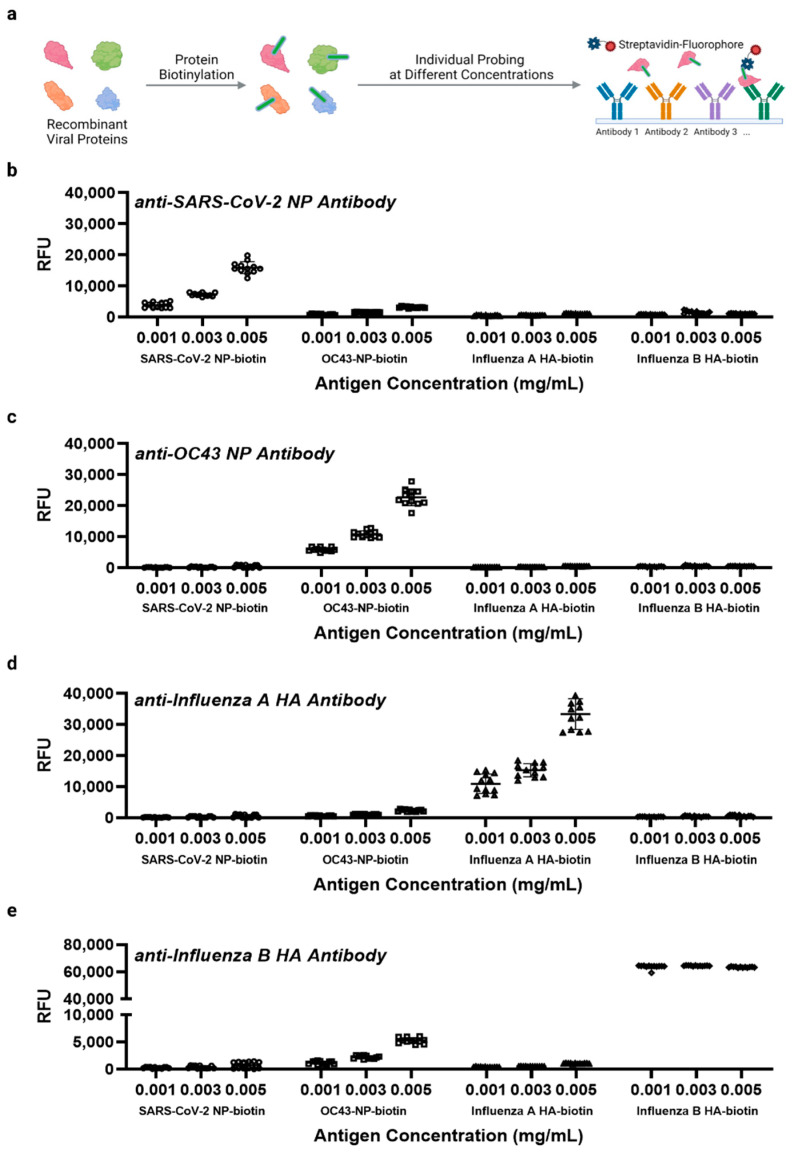
Detection of four recombinant viral proteins after biotinylation using the antibody microarray. On each pad, four different capture antibodies were printed with six replicates each. (**a**) Protein biotinylation and sample probing are illustrated. This panel was created with BioRender.com. Essentially, the detection scheme is identical to Figure 1e. Here, the graphs indicate RFUs from four biotinylated viral antigens detected at three different concentrations by specific capture antibodies: (**b**) anti-SARS-CoV-2 NP (reagent number 14 Appendix A), (**c**) anti-OC43 NP (reagent number 18 Appendix A), (**d**) anti-influenza A hemagglutinin (HA) (reagent number 16 Appendix A), and (**e**) anti-influenza B HA (reagent number 17 Appendix A). Two independent experiments were performed.

We next tested the multiplex viral antigen detection with the heat-inactivated SARS-CoV-2 sample and chemically inactivated influenza A virus sample. These inactivated viruses were lysed, and total proteins were biotinylated for probing (Figure 6a, see Section 2 for details). We demonstrated that the multiplexed antibody microarray can selectively detect the corresponding target viral antigens (Figure 6b,c). However, we observed some cross-reactivity between anti-Influenza B HA antibody and biotinylated inactivated SARS-CoV-2 sample (Figure 6b). Accordingly, the influenza B antigen (HA) cannot be well distinguished, compared to the other targets such as SARS-CoV-2 NP and Influenza A HA. Such non-specificity should be resulted from the anti-influenza B HA antibody, not from the assay design. Future work will need to identify a more specific anti-Influenza B HA antibody. We also noticed that this total sample biotinylation approach introduced a higher fluorescence background compared to the no antigen negative control; nevertheless, we included the IgG negative control printed on the same array (Figure 6b,c) that can be used to normalize the signal obtained from true target binding, if needed.

**Figure 6 biomedicines-10-02238-f006:**
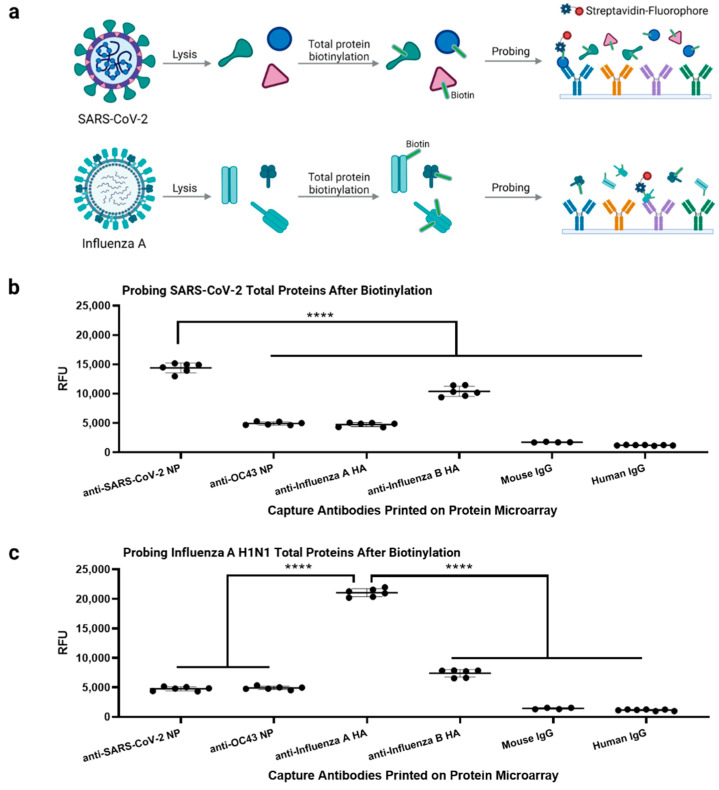
Multiplex detection of viral antigens from inactivated SARS-CoV-2 and influenza A virus after total protein biotinylation. (**a**) To detect the target antigens (SARS-CoV-2 NP and Influenza A HA), inactivated viruses were lysed, and total proteins were biotinylated for probing. This panel was created with BioRender.com. (**b**) Detection of inactivated SARS-CoV-2 sample using the protein microarray is shown. (**c**) Detection of inactivated influenza A H1N1 sample using the protein microarray is shown. The anti-Influenza B HA antibody resulted in the highest non-specific signals. One-way ANOVA, followed by Tukey’s multiple comparison test, was performed for statistical analysis. A single experiment with six technical replicates were tested for each antibody, except the mouse IgG (four technical replicates). **** *p* < 0.0001.

## 4. Discussion

In this study, we demonstrate the iterative development process one can follow to develop protein microarray tests for multiplexed viral antigen detections. Specifically, our antibody microarray-based SARS-CoV-2 antigen test exhibited the clinical sensitivity and specificity of 77.2% and 100%, respectively, which are comparable to existing SARS-CoV-2 antigen tests [34,44]. Of note, we demonstrate the microarray assay correlates well with RT-PCR Ct values and is particularly effective in identifying high viral titers (i.e., low Ct values) that have shown to associate with infectivity and transmission [39,48,49]. However, although our clinical sensitivity matches to some of literature values, such as other antigen tests, our assay design would be challenging to detect early and asymptomatic COVID-19 patients [7].

Antigen-based SARS-CoV-2 tests generally have a lower sensitivity than RT-PCR based methods [43,50], but unlike most of antigen tests, high-throughput and rapid protein microarray technology with a portable imager can be an effective screening tool to identify patient carriers of viable and high viral loads of SARS-CoV-2 for community surveillance and mitigation to aid quarantine process. Other advantages of the protein microarray are the stability and storage conditions. Although we did not test a long-term shelf life of all versions of the protein microarrays used in this study, the same batch of microarrays prepared for the clinical sample testing were used from 15 June 2021 to 6 October 2021. Within this period, we did not find any issues regarding data reproducibility. We expect our low-cost microarray test is readily amenable for automation, scale-up, and deployment to meet demand of COVID-19 surveillance. In future work, for instance, we will transition our viral antigen profiling microarray from a standard microscope slide format (16 arrays/slide) to a microtiter plate format (96 arrays/plate) in order to leverage the ubiquity of clinical laboratory automation equipment designed around this format. 

As we will likely continue to have the COVID-19 pandemic, seasonal common cold and flu epidemics at the same time in the future, our ability to analyze the multi-respiratory viral panel is therefore useful and advantageous, compared to existing methods (e.g., RT-PCR, LFA) that typically only detect one or two targets in discriminating these viral infections and identifying the cause of infection when they share similar symptoms. Our future microarray iterations will incorporate additional antibodies to target a comprehensive panel of respiratory viruses including additional strains of influenza viruses, coronaviruses, adenoviruses, parainfluenza viruses, metapneumoviruses, and respiratory syncytial viruses. Each microarray test can accommodate 100 s of different antibodies including those that target different antigen epitopes from the same virus, which allows us to further increase sensitivity and specificity. In particular, cross-reactivity of antibodies to closely related viruses can be addressed via multiple antigen receiver operating character (ROC) analysis, as shown in our previous COVAM assay [18].

A key advantage of microarray platform is its capability to screen numerous target biomarkers, reagents (e.g., antibodies) and experimental conditions in a high-throughput manner [27,51], greatly reducing assay development time for emerging pathogens. Therefore, the microarray assay can be used in conjunction with other assay platforms for diagnostic assay development [52] where microarray is used for biomarker discovery and reagent screening. Then, downstream assays (e.g., RT-PCR, ELISA, LFA or bead-based immunoassays) can incorporate a small subset of biomarkers/antibodies for final diagnostic tests. As a result, this protein microarray platform can be broadly applied for combating COVID-19, quantitative characterization of virologic and immunologic responses and highly parallel antibody/antigen binding in basic research, and supporting the development of diagnostics, therapeutics, and vaccines. Furthermore, the microarray panel can be easily customized to meet the needs of many other human pathogenic viruses (e.g., Human immunodeficiency virus, Zika virus, Dengue virus, and Ebola virus) and future emerging pandemic threats beyond the current COVID-19 pandemic.

## 5. Conclusions

As COVID-19 will prevail and as well as to prepare for emerging pandemics in the future, rapid and accurate diagnostic tests are needed. Antigen test, which may suffer from its suboptimal sensitivity compared to molecular tests, can play its unique role to fight against pandemics. Here, we present a general workflow to develop and test both singleplex and multiplex antigen detection platform using antibody microarray technology, that is coupled with a portable imager. Antibody microarray can function not only as a research tool to screen multiple antibodies and reagents to aid in developing other immunoassay formats such as LFA but also as a diagnostic tool. Further improvements in this technology, including but not limited to automation of the sample preparation and data analysis, can be applied in the future.

## Figures and Tables

**Figure 2 biomedicines-10-02238-f002:**
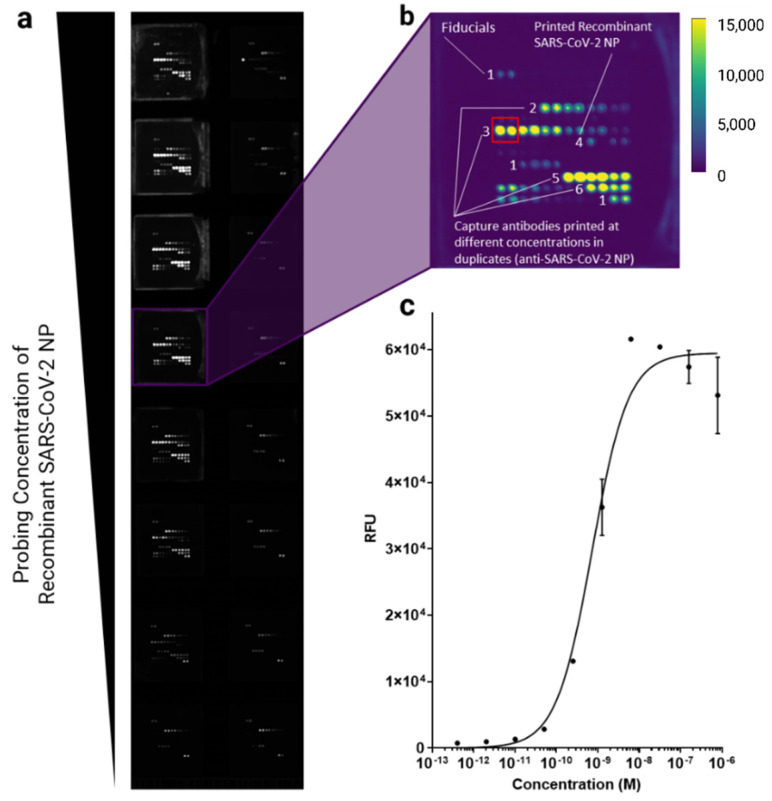
Representative antibody pair screening experiment. Recombinant SARS-CoV-2 NP was serially diluted and probed on pads. (**a**) A representative protein microarray image showing all 16 pads which are consisted of identical arrays of proteins. The brightness of the image was adjusted from 0 to 15,000 using ImageJ. (**b**) A representative image of a single pad is shown with labels. Each label indicates the following reagents: label 1—fiducials, label 2, 3, 5, and 6—anti-SARS-CoV-2 NP capture antibodies (reagent number 8, 14, 11, and 12 in Appendix A, respectively), and label 4—recombinant SARS-CoV-2 NP (reagent number 1 in Appendix A). The anti-SARS-CoV-2 NP capture antibodies were printed at different concentrations in duplicates. Fluorescence signals captured from the highest printing concentrations (red box) were used from each pad to calculate K_D_ and Bmax. (**c**) A representative graph after curve-fitting using the antibody pair 1 in Appendix A is shown. Recombinant SARS-CoV-2 NP was serially diluted in a five-fold and probed on protein microarray. A total of ten concentration points were tested. After probing, fluorescence signals from each concentration point were measured and recorded. Using a built-in function in the PRISM software (nonlinear regression, one site—specific binding), K_D_ and Bmax were calculated, which were used for the main parameters to determine the best antibody pair to detect SARS-CoV-2 NP on the protein microarray. Some error bars were not present on the graph because they were too small to be drawn. A single experiment with two technical replicates was performed for each antibody pair.

## Data Availability

Not applicable.

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
