# Peer review of "A Protein Microarray-Based Respiratory Viral Antigen Testing Platform for COVID-19 Surveillance"

_biomedicines, 2022, doi:10.3390/biomedicines10092238_

Round 1

Reviewer 1 Report

This is an interesting manuscript in which the authors describe the respiratory viral antigen testing Using protein microarrays. The manuscript is well written. However, I consider this manuscript of interest but after moderate revision. 

1.     It should indicate how much of the arrayed antibody is capable of binding to the antigen.

2.     The antibody-immobilized arrays are stored in a dry state, but does this reduce the binding activity of the antibodies?

Author Response

Point 1: It should indicate how much of the arrayed antibody is capable of binding to the antigen.

Response: We partially addressed this comment in the original manuscript, line 702 - 704: "that each microarray test can accommodate 100s of different antibodies including those that target different antigen epitopes". However, to further address this comment, we agree to add more descriptions about antibody printing on nitrocellulose pads. Theoretically, each specific antibody printed on a nitrocellulose pad can capture its target antigen. Our experiment showed (results not shown here) that deposited monoclonal antibodies on the arrays interact with specific antigen at stoichiometric ratio 1:1. Nevertheless, multiple factors such as KD (dissociate constant) and therefore printing concentration of each antibody can affect the efficiency of antigen binding. Typically, 16-pads slide can accommodate about 250 spots per pads whereas 4-pads slide can accommodate about 2000 spots per pad.

Point 2: The antibody-immobilized arrays are stored in a dry state, but does this reduce the binding activity of the antibodies?

Response: Our multiple experiments showed that proteins on microarrays (including IgGs and anti-IgGs) do not reduce binding activity at least for slide expiration period (2 years). The storage condition is at low humidity in a desiccator at 23 - 25C. Concerning this experiment, we did not test a long-term shelf life of the particular protein microarrays, but the same batch of microarrays prepared for clinical sample testing were used from 2021 June 15 to 2021 October 6. Within this period, we did not find any issues regarding data reproducibility. This information can be further discussed and added in the line 689 – 690 in the original manuscript. 

Reviewer 2 Report

Review of the manuscript entitled “A protein microarray-based respiratory viral antigen testing platform for COVID-19 surveillance”

The article aims to develop a protein-based microarray assay for the detection of respiratory viruses. The authorsdemonstrated that the antibody microarray assay exhibited a clinical sensitivity and specificity of 77.2% and 100%, respectively, which are comparable to existing FDA-authorized antigen tests. The multiplexed assay can selectively detect SARS-CoV-2 and influenza virus, which can be used to discriminate these viral infections that both share similar symptoms. A microarray TinyArray imager was used for protein imaging and data analysis. Descriptions of methods are accurate and potentially able to be repeated. The manuscript is suitable for the journal and eventually can be accepted after some minor improvements:

Comment for the authors:

A brief rationale for selecting a nucleocapsid protein for further investigation would be desirable.

The rationale for buffer selection is confusing. The text should be more consistent, and the most important information should be presented in such a way that it is not necessary to search for it.

The abstract could be extended with some more details about the methods used.

The introduction is rather limited, related references from MDPI journals are not taken into account. In this research immunosensor for SARS-CoV-2 antigen detection is reported, therefore, some additional references on the determination of SARS-CoV-2 antigens or antibodies against these antigens by some other methods could be taken into account. Some electrochemical sensors with artificial specific binding sites i.e., molecularly imprinted polymers could be discussed in the introduction, and sensitivity and other aspects could be compared. Some reviews on the development of biosensors for the determination of antibodies against SARS-CoV-2 Virus proteins could be taken into account. 

The sentence is split by figure: Initial part of the sentence at Line 507, while the of the sentence at Line 528.

In figure 1, some schematic symbols in drawings are not explained and are not clear, particularly the biotinylation, etc. Please check and explain all these symbols used in the drawings. In figure 1, part c, the line towards antigen is drawn not very exactly, please correct the direction of this line/arrow.

In figure 2c, some error bars are not present on the graph, because they were too small to be drawn, however, some of these points are far from presented sigmoidal dependence, therefore some additional explanation/discussion regarding this kind of accuracy could be added to related Discussion.

Line 259-262, check the correct introduction of ‘RFUs’ - relative fluorescence units, they are mentioned several times in these sentences. It should be noted that abbreviations should be clearly introduced before first use in the text.

The final conclusion could be extracted and presented after the discussion part.

Author Response

Point 1: A brief rationale for selecting a nucleocapsid protein for further investigation would be desirable.

Response: The rationale of selecting SARS-CoV-2 nucleocapsid protein (NP) as the target analyte over spike protein was because NP is approximately 10-fold more abundant; therefore, NP was expected to provide high sensitivity for our immunoassay. This was mentioned in the line 394-398 (in the result section) of the original manuscript: “Bar-On, Y.M.; Flamholz, A.; Phillips, R.; Milo, R. Science Forum: SARS-CoV-2 (COVID-19) by the numbers. elife 2020, 9, e57309”. However, we agree that we can add further description in the introduction section.

Point 2: The rationale for buffer selection is confusing.

Response: As mentioned in the Materials and Methods section, four different lysis buffers were tested. Out of the four buffers, LY-13 buffer was determined to yield optimal signal-to-noise ratio with the least variation and, therefore, subsequently used for assaying heat inactivated SARS-CoV-2 and all clinical samples. (Also see Figure S1 for experimental data).

Point 3: The text should be more consistent, and the most important information should be presented in such a way that it is not necessary to search for it.

Response: We agree with the reviewer's comment. We will address this in the revised manuscript.

Point 4: The abstract could be extended with some more details about the methods used.

Response: Abstract space limitation does not allow us to go into more methodical details. Therefore, to be more specific, original “protein-based microarrays” will be replaced with “antibody-based microarrays”.

Point 5: The introduction is rather limited, related references from MDPI journals are not taken into account.

Point 5_1: Some additional references on the determination of SARS-CoV-2 antigens or antibodies against these antigens by some other methods could be taken into account.

Response: 

We agree with the reviewer. To address this comment, we will cite two published papers from the MPID Biomedicines, specifically published for this special issue of High Sensitivity Lateral Flow Assays for SARS-CoV-2 and Other Infections:

Thapa, Damber, et al. "Rapid and Low-Cost Detection and Quantification of SARS-CoV-2 Antibody Titers of ICU Patients with Respiratory Deterioration Using a Handheld Thermo-Photonic Device." Biomedicines6 (2022): 1424.

Giberti, Irene, et al. "High Diagnostic Accuracy of a Novel Lateral Flow Assay for the Point-of-Care Detection of SARS-CoV-2." Biomedicines7 (2022): 1558.

Point 5_2: Some reviews on the development of biosensors for the determination of antibodies against SARS-CoV-2 Virus proteins could be taken into account.

Response: 

We agree with the reviewer. To address this comment, we will cite a following review article form the MDPI Biomedicines, which discusses monoclonal antibodies and their use for viral infections:

Mokhtary, Pardis, et al. "Recent Progress in the Discovery and Development of Monoclonal Antibodies against Viral Infections." Biomedicines 10.8 (2022): 1861.

Point 6: The sentence is split by figure: Initial part of the sentence at Line 507, while the of the sentence at Line 528.

Response: We checked the line 507 - 528. However, we could not find the sentence split

Point 7: In figure 1, part c, the line towards antigen is drawn not very exactly, please correct the direction of this line/arrow.

Response: We agree with the reviewer. Thank you for spotting the line. We will explain the other schematic symbols in drawings with more details in the revised manuscript.

Point 8: In figure 2c, some error bars are not present on the graph, because they were too small to be drawn, however, some of these points are far from presented sigmoidal dependence, therefore some additional explanation/discussion regarding this kind of accuracy could be added to related Discussion.

Response: 

We believe such decrease in the RFU from the highest analyte concentration point may have been resulted from high-dose hook effect, where too much analyte saturated both capture and detection antibodies, yielding low signal. We can also re-cite a reference which discusses the high-dose hook effect in detail:

Wild, David, ed. The immunoassay handbook: theory and applications of ligand binding, ELISA and related techniques. Newnes, 2013.

Point 9: Line 259-262, check the correct introduction of ‘RFUs’ - relative fluorescence units, they are mentioned several times in these sentences. It should be noted that abbreviations should be clearly introduced before first use in the text.

Response: The sentence “Negative RFUs (relative fluorescence units) were converted to zero….” will be corrected to: “Negative relative fluorescence units (RFUs ) were converted to zero…”.

Comment 10: The final conclusion could be extracted and presented after the discussion part.

Response: We agree with the reviewer. 

Reviewer 3 Report

An antigen-based platform for sensing SARS-CoV-2 virus has been presented. The subject is interesting, important and publishable. But, there are some points which should be clarified, addressed and/or discussed in the revised version, as mentioned below:

11.       1.       Some of the recent antigen testing of SARS-CoV-2 can be found in [CaZnO-based nanoghosts for the detection of ssDNA, pCRISPR and recombinant SARS-CoV-2 spike antigen and targeted delivery of doxorubicin] & [https://doi.org/10.1002/14651858.CD013705.pub2]. This should be mentioned in the introduction section.

22.       The LOD of this work should be compared with the LOD of the previous works in the literature. A table should be designed for this task. Whether can this LOD be considered as a suitable index for early detection of SARS-CoV-2? Please discuss using suitable supports.

33.       The sensing time (2.5 h) should be compared with other assays. It does not seem so online. Please discuss in this regard further.

44.       Figure 6 shows that the designed biosensor cannot distinguish SARS-CoV-2 from influenza BH1. So, this cannot be considered as selective sensor. This should be clearly explained in the revised version.   

55.       Testing and also maintaining the appropriate level of oxygen in the blood circulation are known as important parameters in controlling Covid-19 (see, for example, [ACS Applied Nano Materials 4 (2021) 11386-11412]). This should be mentioned in the introduction section.  

66.       The statistics of the measurements (such as P-values and the number of experiments) should be given (even in the captions of the related figures).

77.       Some recent review regarding the antigen-based sensing of Covid-19 can be found in [ACS Biomaterials Science & Engineering 8 (2021) 54-81] & [ACS Infect. Dis. 2020, 6, 8, 1998–2016].  These progresses should be mentioned in the revised version for further completion of the literature review.

88.       It is useful that the authors give a real optical or microscopical image from the assay in Figure 1. No evidence relating the microstructure can be found in the manuscript.

Author Response

Point 1: Some of the recent antigen testing of SARS-CoV-2 can be found in [CaZnO-based nanoghosts for the detection of ssDNA, pCRISPR and recombinant SARS-CoV-2 spike antigen and targeted delivery of doxorubicin] & [https://doi.org/10.1002/14651858.CD013705.pub2]. This should be mentioned in the introduction section.

Response: We agree with the reviewer's comment. We will cite these suggested references and discuss further in the introduction section.

Point 2: Testing and also maintaining the appropriate level of oxygen in the blood circulation are known as important parameters in controlling Covid-19 (see, for example, [ACS Applied Nano Materials 4 (2021) 11386-11412]). This should be mentioned in the introduction section.

Response: We agree with the reviewer's comment. We will cite these suggested references and discuss further in the introduction section.

Point 3: Some recent review regarding the antigen-based sensing of Covid-19 can be found in [ACS Biomaterials Science & Engineering 8 (2021) 54-81] & [ACS Infect. Dis. 2020, 6, 8, 1998–2016]. These progresses should be mentioned in the revised version for further completion of the literature review.

Response: We agree with the reviewer's comment. We will cite these suggested references and discuss further in the introduction section.

Point 4: The LOD of this work should be compared with the LOD of the previous works in the literature. A table should be designed for this task. Whether can this LOD be considered as a suitable index for early detection of SARS-CoV-2? Please discuss using suitable supports.

Response: 

As discussed above, we will cite more key references regarding the SARS-CoV-2 antigen tests and their sensitivity (LOD). However, due to the massive volume of literature information, we will discuss only the most relevant information in the revised manuscript: the LOD of fluorescence-based antigen test discussed in the suggested review article:

Dinnes, Jacqueline, et al. "Rapid, point‐of‐care antigen tests for diagnosis of SARS‐CoV‐2 infection." The Cochrane Database of Systematic Reviews 2021.3 (2021).

Point 5: The sensing time (2.5 h) should be compared with other assays. It does not seem so online. Please discuss in this regard further.

Response: We agree with the reviewer's comment. The protein microarray test discussed in this manuscript does not follow a typical LFA protocol. Rather, it follows Fluorescence-linked immunosorbent assay (FLISA) format, where the majority of the assay time is resulted from incubation time for each step.

Point 6: Figure 6 shows that the designed biosensor cannot distinguish SARS-CoV-2 from influenza BH1. So, this cannot be considered as selective sensor. This should be clearly explained in the revised version

Response: We agree with the reviewer's comment. The influenza B antigen (HA) cannot be well distinguished, compared to the other targets such as SARS-CoV-2 NP and Influenza A HA. However, this non-specificity is resulted from the anti-influenza B HA antibody, not from the assay design. Although we spent significant amount of time and effort to screen right and suitable reagents, it is challenging to find an optimal antibody for specific analyte. Accordingly, this re-addresses one of our research focuses that is the protein microarray technology can provide such a platform to screen multiple antibodies, reducing time and labor for researchers. Nevertheless, this can be further discussed in the discussion section. 

Point 7: The statistics of the measurements (such as P-values and the number of experiments) should be given (even in the captions of the related figures).

Response: We agree with the reviewer's comment. We will mention the p-values and the number of experiments in the figure legends if missed.

Point 8: It is useful that the authors give a real optical or microscopical image from the assay in Figure 1. No evidence relating the microstructure can be found in the manuscript.

Response: We presented a raw image of a typical 16-pad protein microarray chip using the TinyImager (the portable imager developed for rapid imaging acquisition) after sample probing in the figure 2a. Using a built-in function provided from the ImageJ software, we specifically color-coded and adjusted brightness of a single pad with annotations in the figure 2b.

Round 2

Reviewer 3 Report

The manuscript has been revised based on the comments and now is publishable.